# Checkpoint Inhibitors and Hepatotoxicity

**DOI:** 10.3390/biomedicines9020101

**Published:** 2021-01-21

**Authors:** Stephen D. H. Malnick, Ali Abdullah, Manuela G. Neuman

**Affiliations:** 1Department of Internal Medicine C, Kaplan Medical Center, Rehovot 76100, Israel; aliah@clalit.org.il; 2In Vitro Drug Safety and Biotechnology, Toronto, ON M5G 1L5, Canada; m_neuman@rogers.com; 3Department of Pharmacology and Toxicology, Faculty of Medicine, University of Toronto, Toronto, ON M5G 1L5, Canada

**Keywords:** checkpoint inhibitors, hepatotoxicity, drug induced liver injury, tumor cells, immunological escape, malignant cells, cytokines

## Abstract

Uncontrolled immune response to a pathogen or any protein can lead to tissue damage and autoimmune diseases, that represent aberrant immune responses of the individual to its own cells and/or proteins. The immune checkpoint system is the regulatory mechanism that controls immune responses. Tumor cells escape the immune surveillance mechanism, avoiding immune detection and elimination by activating these checkpoints and suppressing the anti-tumor response, thus allowing formation of tumors. Antigenic modulation facilitates masking and contributes to the escape of tumor cells. In addition, there are growing cell promoters, like transforming growth factor β (TGF-β), contributing to escape mechanisms. Targeting the immunological escape of malignant cells is the basis of immune oncology. Checkpoint inhibitors, cytokines and their antibodies may enhance the immune system’s response to tumors. Currently, immunomodulatory agents have been designed, evaluated in clinical trials and have been approved by both European and United States Drug Agencies. The present review is a reflection of the increasingly important role of the checkpoint inhibitors. Our aim is to review the side effects with the emphasis on hepatic adverse reactions of these novel biological drug interventions.

## 1. Introduction

Immune checkpoint inhibitors (ICIs) are mainly antibodies with an immunomodulatory function [1]. It is thought that by stimulating the immune system there is an improved prognosis for patients with advanced stage malignancies.

Both the innate and adaptive immune systems play an important role in the immune surveillance of cancer. This enables malignant cells to be detected and eliminated at early stages. Initially, there is an elimination phase which consists of the innate and adaptive response to specific tumor-associated antigens. Next, there is an equilibrium phase which involves a balance between immune-mediated destruction by the adaptive immune system and the persistence of rare malignant clones. Subsequently, there is immunological escape, whereby malignant clones have acquired the ability to invade the adaptive immune system.

## 2. Tumor Microenvironment

The tumor microenvironment (TME) is intimately involved with the regional immune response. Both the interaction with the intrinsic and extrinsic components of the tumor, together with the cytokines and chemokines in the TME, aid in the development of angiogenic aggressiveness of the tumor. Cytokines communicate with the immune system and represent direct intercellular contact between cancer cells and the tumor microenvironment [2].

Cancer immunotherapies use ICI to stimulate the immune systems of patients in order to induce anti-tumor activities. The immune checkpoint blockade with antibodies ICIs in physiologic conditions functions as a regulator of excessive inflammation. Cancer cells upregulate the expression of immune checkpoint molecules. The ligands of these molecules inactivate T cells. Thus, the mechanism of action of ICIs is the reactivation of T cells, aiming to apply cytotoxic activities over cancer cells.

Regulatory T cells (Tregs) impair effector T cell function. There are several mechanisms of Treg-mediated immune suppression, including direct cell-to-cell contact. The production and export of powerful immunosuppressive cytokines including interleukin (IL)-10, IL-35 and transforming growth factor β (TGF-β) will result in the differentiation of naïve T cells into inducible Tregs [3].

Tumor cells promote cancer progression by secreting growth factors, which stimulate cancer cell growth and the formation of tumor blood vessels. Important factors in this process include cytokines tumor necrosis factor alpha (TNF-α), chemokines and several growth factors, including vascular endothelial growth factor (VEGF), platelet-derived growth factor (PDGF), epidermal growth factor (EGF), fibroblast growth factor (FGF) and TGF-β. Prolonged production of these factors by tumor-supporting immune cells accelerates cancer cell growth, together with the development of tumor blood vessels. The adaptive immune system modulates the maturation of dendritic cells (DC) and their activation via nuclear killing (NK) cells by killing tumor cells and releasing tumor antigens: NK cells stimulate major histocompatibility complex (MHC) class I context. NK cells also prime DCs to release IL-12. Interleukin 12 activity is necessary for functional T helper 1 cells (Th1), resulting in the production of IL-21, which in turn is involved in reciprocal stimulation of NK cells.

The interaction between the extracellular matrix (ECM), immune cells, blood vessels formation and the inflammatory response is part of the TME. The high levels of cytokines and chemokines in the TME amplify the tumor-related angiogenesis. Pathways of communication involving cytokines, together with direct cell contact of tumor cells and the TME, contribute to the mechanism of tumor aggression.

Targeted negative regulators of immune cell functions are used as immuno- therapies. Blocking the T cell inhibitory signal results in an active immune response leading to tumor cell death. Maintenance of T cell activation by stimulatory and inhibitory regulators prevent autoimmune inflammation or immune deficiency.

The anti-tumor immune response is over-expressed. Receptors on the T cell surface cause the immune cells’ deactivation, resulting in the progression of the cancer. Some of the targeted genes are lymphocyte activation gene 3 (LAG-3) and T-cell immunoglobulin and mucin-domain containing-3 membrane protein (TIM-3). LAG-3 is a membrane protein that binds MHC-II, similarly to CD4 [4,5]. This results in T cell suppression and a decrease in cytokine release and is also implicated in T cell exhaustion in both viral infection and cancer [6,7]. The combination of LAG-3 and PD-1 has been shown to synergistically promote tumor regression [8].

T-cell immunoglobulin and mucin-domain containing-3 membrane protein (TIM-3) is known to be expressed on CD4+ T Helper 1 cells and CD8+ cytotoxic cells. [9,10]. TIM-3 enables tumor cells to evade immuno-surveillance. Some CD8+ cells co-express both PD-1 and TIM-3, adding to the dysfunctional CD8+ T cells [11]. Blocking both TIM-3 and PD-1 produces double immuno-surveillance [12].

## 3. Biomarkers for Severity of the Disease

There is a need to employ specific biochemical and pathological unbiased differential indicators of illness onset, in order to assist in the classification of a diseased or non-diseased state. Biomarkers provide the ability to stage disease progression and its severity. Such prognostic indicators could be used for risk stratification of populations. In addition, biomarkers can assist in the recognition of the efficacy of clinical or therapeutic interventions of disorders.

It is hoped that the clinician will be able to choose the optimal therapy based on biomarker testing to identify potential actionable gene aberrations. Therefore, it is necessary to employ either specific serum biomarkers or tissue biopsy to identify mutations. The biomarkers are utilized to classify candidates who could benefit from selective check-point inhibition as well as monitoring the response to therapy.

Biomarkers assist in the classification of gene expression profiling of the tissue. Gene expression profiling attempts to identify the mutational or phenotypic profile specific to the tumor. By investigating the molecular genetics of tumors, it is possible to use targeted therapies. However, some studies did not reveal a significant difference in gene expression between specific and non-specific tumors [13].

For example, amplification of oncogene protein c-MYC, protein IK3 oncogene (PIK3CA), human estrogen receptor (HER)2 and fibroblast growth factor signaling pathway (FGFR1), and mutation of suppressor protein (p53), breast cancer susceptibility alleles (BRCA)2 and tumor suppressor gene with protein and phosphatase activity (PTEN), were detected in individuals with inflammatory breast cancer (IBC). However, only 42% of patients with triple-negative breast cancer had MYC amplification. As a result, further studies are required in order to elucidate the molecular profile and improve the survival rate of patients [14].

Small non-coding RNAs, microRNAs (miRNAs), have also been actively investigated as molecular biomarkers for the diagnosis and prognosis of tumors. miRNAs influence the tumor’s regulation of gene expression by targeting messenger RNA (mRNA)s. Qi et al. described five potential miRNAs as diagnostic molecular biomarkers in IBC (miR-301b, miR-451, miR-15a, miR-342-3p and miR-342-5p), some associated with a better (miR-19a, miR-7, miR-324-5p) and others with a poorer prognosis (miR-21, miR-205) [15]. Thus, information of the tissue characteristic is important in therapeutic management.

Other studies described a lower expression of miR-26b in breast cancer compared to normal breast tissue and lower expression of miR-205 in inflammatory breast cancer compared with the non-inflammatory type. Lower expression of both miR-26b and miR-205 was associated with shorter distant metastasis-free survival and overall survival [16].

The potential application of miRNAs for diagnosis and prognosis requires further studies. There is a need to identify differences in the gene expression of cancers to find out the targetable genomic drivers. These factors need to be considered before prescribing an ICI. The response of the tumor to a specific inhibitor is also a function of the surrounding healthy cells.

Figure 1 illustrates the microenvironment, in which the liver cells are exposed to several noxious agents and signals. The immune check point inhibitors assist various types of cells in different organs to commit suicide (apoptosis) or to become necrotic.

Pathogenesis and behavior of the tumors are related to tumors surrounded by inflammatory responses and immune cells, blood vessels and ECM, all of which are components of the TME. The TME has a crucial role in the local immune response. The intimate interaction between the immune systems intrinsic and extrinsic components together with a high concentration of cytokines and chemokines in the TME increase both the aggressiveness and angiogenic potential of tumors. Cytokine-mediated communication and direct intercellular contact between cancer cells and TME with a variety of pathway crosstalk are critical means of interaction. For example, liver cells secrete integrins, which form a complex with cell adhesion molecules and vascular cell adhesion molecule-1 that suppresses apoptosis in metastatic breast cancer cells [17,18]. In Figure 2, we present only the reaction of the parenchymal cell (hepatocyte) to the ICI.

The different microbiota together with reactive oxygen species (ROS) may lead to an increased uptake of endotoxins. These endotoxins—lipopolysaccharides (LPS)—are part of the activation of nuclear factor kB (NF-kB) and assist in the release of various proinflammatory cytokines (Il-8, Il-6, TNF, Il-1ß) and chemokines, including CC-chemokine ligand 2 (CCL2). Together, these factors sensitize the hepatocytes. The survival factors are reduced and there is mitochondrial damage that can be observed by reduction of the specific enzyme succinate dehydrogenase (SDH) and release of cytochrome c. ROS bind to proteins, reducing the proteosome, changing their functional and structural properties, and generate neoantigens. In addition, ROS bind directly to and damage DNA. The relationship between ICI and cell damage and death of hepatocyte is shown in Figure 2.

In addition, cytokines can induce synergistic effects on NK cells’ effector functions. It has been reported that IL-12 increases IFN-γ production in IL-15-stimulated NK cells. The tumor microenvironment supports growth of the cancer cells within the tumor. There is a complex relationship between the tumor cells and the body’s immune cells. The cell is primed or sensitized via several environmental factors that may lead to cell death or cell damage. The secretion of LPS by microbiota, ROS, loss of glutathione reserve, mitochondrial toxicity (SDH) and immune events, are producing an increase in pro-inflammatory cytokines.

Multiple growth factors and cytokines may enhance anti-angiogenic pathways promoting pro-angiogenic signaling, such as vascular endothelial growth factors (VEGF).

Lipopolysaccharides are toxins produced by microbiota that reach the liver via the portal vein. In addition, reactive oxygen species (ROS) may sensitize the hepatocytes, reducing succinate dehydrogenase (SDH) activity and releasing cytochrome c. This increases inflammation and assists in cell death by apoptosis or necrosis. The resulting cytokine storm leads to inflammation, cell sensitization and liver cell death.

## 4. Checkpoint Inhibitors

Malignant tumors, however, can avoid or actively suppress the anticancer immune responses. Checkpoint inhibitors are immunomodulatory antibodies that are used to enhance the immune response and have resulted in an improved prognosis for patients with advanced malignancy. There are two main targets for checkpoint inhibition [1]. The first is programmed cell death receptor 1 (PD-1) and programmed cell death ligand 1 (PDL-1). PD-1 is involved in NK cell collapse, limiting their toxic activity and cytokines production. The additional target is cytotoxic T-lymphocytes-associated antigen 4 (CTLA-4). CTLA-4 is implicated in the inhibition of IFN-γ assembly by NK cells induced by DCs.

Checkpoint inhibitors, perhaps because of their marked effect on the immune system, are associated with a unique spectrum of side effects [19,20]. These are termed immune-related adverse effects. It is the purpose of this review to discuss the side effects of these novel therapies with emphasis on hepatocytotoxicity.

## 5. Approved Checkpoint Inhibitors

There are several ICIs approved for cancer therapy [21]. Ipilimumab is an anti-CTLA-4 antibody, it is used for treatment of patients with advanced-stage melanoma. Tremelimumab is another anti-CTLA-4 antibody that is under development.

There are several antibodies against PD-1 and PD-L1. Nivolumab, pembolizumab and cemiplimab target PD-1 and atezolizumab, while avelumab and durvalumab target PD-L1. These antibodies have been approved for several indications, including hepatocellular carcinoma, melanoma, renal cell carcinoma, bladder cancer, non-small cell lung cancer, head and neck cancer urothelial carcinoma, Hodgkin’s lymphoma, Merkel cell carcinoma as well as in solid tumors associated with microsatellite instability and high or mismatch repair deficiency (dMMR). The safety and activity of ICIs in patients with other morbidities, including autoimmune disorders, organ transplants, patients with chronic viral infections, patients with long-term immunosuppressant use, or presenting organ dysfunction, pregnant women, patients with brain metastases, pediatric or aged patients, as well as patients with an impaired functional status, is of higher concern [21].

## 6. Hepatic Side Effects

As might be expected, ICIs that have such an extensive effect on the immune system can have deleterious effects. Checkpoint molecules are expressed in T-cell populations that have specificity for self-antigens. The non-specific activation of the immune system associated with ICIs may result in side-effects in many organs [22]. The hyper-activation of the T-lymphocytes generates on-target activity against normal tissues in addition to attacking tumor-specific antigens.

CD8+ cytotoxic T-lymphocytes result in the destruction of tumor cells with the release of tumor antigens, neo-antigens and auto-antigens from normal tissues. This is termed epitope-spreading and results in decreased immune tolerance. This effect, together with activation of both Th1 and Th17 T-lymphocytes, results in the production of pro-inflammatory cytokines, including interferon-gamma (IFN-γ) and interleukin-17 (IL-17). In addition to this mechanism, there may be cross-reactivity with the microbiome, hypersensitivity and a specific effect of PDL-2 [23].

The organs that are most commonly affected include the skin, endocrine organs, gastrointestinal tract (GI), liver and lungs. It is the purpose of this review to examine the hepatic adverse effects of the novel class of immune therapy: ICIs.

The subject of ICI-related hepatotoxicity has been the subject of systematic reviews [3,24,25]. The incidence varied from 0.7% to 1.6%. This difference may be related to differences in dosage, the type of ICI and monotherapy versus combination ICI treatment. The incidence is lowest for PD-1, increases for PDF-L1 and is highest for CTLA-4. The highest incidence (13%) results from the combination of CTLA-4/PD-1 (13%) and high-dose CTLA-4 (16%). In addition, the incidence of grade 3–4 hepatotoxicity was 0.6–11%, but highest in those with the highest dose of CTLA-4 inhibitors. There were also cases of fulminant hepatic failure (0.1–0.2%) [26].

The preferred method for determining causality is the Roussel Uclaf Causality Assessment (RUCAM) [27]. The time to onset of liver injury using the RUCAM model has been found to be between 4 and 12 weeks (1–3 courses of chemotherapy). Following cessation of ICI, liver enzyme normalization ranges from 8 to 104 days.

## 7. Histology

The data is limited, perhaps due to a reluctance to perform liver biopsy in patients who are so severely ill. The two main findings in the report by Peeraphadit et al. [25] are an acute hepatitis with lobular inflammation, acidophilic bodies and centrilobular necrosis. There also appears to be different hepatic histology between CTLA-4 and PD-1/PD-L1 inhibitors. CTLA-4 inhibitors tend to cause granulomatous hepatitis. In addition, there are reports of granulomas in other organs related to CTLA-4 inhibitors [14,15,25,28,29]. However, Peeraphatdit et al. [25], de Martin et al., as well as Zen and Yeh [30,31] did not see granulomatous hepatitis in the 13 patients they treated. There is not a specific pattern for anti-PD-1/PD-L1 hepatitis. Lobular hepatitis with centrilobular necrosis and periportal activity have been reported.

Hepatotoxicity related to cancer chemotherapy is graded according to the Common Terminology Criteria for Adverse Events (CTCAE) of the National Cancer Institute [32]. The grade of hepatotoxicity is based on peak abnormalities of alanine amino transferase (ALT), aspartate aminotransferase (AST), gamma- glutamyl transpeptidase (GGT) and bilirubin. The Food and Administration in United States (FDA) employs the following formula to define severe drug induced liver injury (DILI): ALT > 3 times upper limit of normal (ULN) and total bilirubin > 2 times ULN [33]. However, cytotoxicity may be present without an elevation of serum bilirubin and with normal hepatic synthetic function.

There are reports from many clinical trials of liver-related adverse effects related to ICI treatment. Although liver damage may develop at any time, most cases are observed within the first three months of treatment [25]. AST/ALT increase was present in 2–5% of cases and a grade 3–4 increase in 1–4% of cases [18,34] treated with anti-PD-1 agents, and there was a 9–15% rate of liver damage and 4–6% of grade 3–4 toxicity [34,35]. The hepatic side-effects are much higher in those patients treated with anti-CTLA-4 antibodies [18,20,36,37]. Combination treatment of both anti-PD-1 and anti-CTLA4 antibodies resulted in a more severe hepatotoxicity: 17.6–21% for all grades and 8.3–11% for grade 3/4 toxicity in patients treated for melanoma [37]. For anti-PD-1 monotherapy, there was an increase in AST or ALT in between 1.8–6.2% of cases and of grade 3/4 reactions between 1.1–1.8% [27,38].

## 8. Hepatic Injury Related to ICI Treatment

The diagnosis of ICI hepatic injury requires excluding other causes of hepatic injury or pathology. Many cases are diagnosed on the basis of incidental findings of elevated liver enzymes, prior to administration of another dose of ICI. A thorough medical history is important. In addition, use of validated methods to detect causality for cases of drug-induced liver injury (DILI) is helpful. The RUCAM scale is a commonly used tool for assessing the likelihood of DILI [27]. Other causes of hepatotoxicity need to be ruled out. These include viral hepatitis, other viral infections, alcoholic liver disease, non-alcoholic liver disease, ischemic hepatitis, autoimmune hepatitis and liver metastasis.

The diagnosis of liver metastasis as a cause for elevated liver enzymes is common in the patients receiving ICIs for advanced malignancies. A series of 491 patients treated with pembrolizumab found that 14.3% had liver injury and of these, more than half had liver metastases [39]. It should be remembered that non-hepatic causes of transaminases can also occur due to ICI. If the AST level is higher than the ALT, or there is no ALP or bilirubin elevation, then one should consider cardiac involvement, including myocarditis or myositis [40].

The histological picture with anti-PD-1/PD-L1 treatment was a lobular hepatitis. It is unclear why the incidence of hepatotoxicity is higher with anti-CTL4 agents compared to anti-PD1 agents. Six of the patients responded to cessation of the treatment, seven received oral steroids at a dose of 0.5–1.0 mg/kg/day, two needed maintenance steroid therapy at a dose of 0.2 mg/kg/day, while one required pulse treatment at a dose of 2.5 mg/kg/day as well as a second immunosuppressive drug. In three patients, reintroduction of the immunotherapy was successful with no further liver dysfunction. Combination therapy with ipilimumab and nivolumab has been reported to result in fibrin ring granulomas in two cases [41].

ICI-mediated hepatotoxicity must be distinguished from autoimmune hepatitis [31]. ICI-mediated hepatotoxicity does not have a female predominance, only 50% have anti-nuclear antibodies and there is less of a plasmacytosis on liver biopsy. In addition, with autoimmune hepatitis, there is much more likely to be a recurrence after withdrawal of steroid therapy (summarized in Table 1). Table 2 summarizes the reported hepatic side-effects of ICI treatment. 

A combination of history of medication, laboratory tests and performed biopsy will usually enable the distinction to be made.

## 9. IPI Treatment of Hepatocellular Carcinoma

Hepatocellular carcinoma (HCC) usually arises in the setting of cirrhosis from dysplastic nodules [56]. There are many causes of cirrhosis, including hepatitis B virus infection (HBV), hepatitis C virus infection (HCV) and non-alcoholic fatty liver disease (NAFLD). HCC is the fourth most-common cause of cancer-related death worldwide. The main elements of treatment of HCC include prevention by successful treatment of both HBV and HCV infection and screening for early lesions. Early disease is treated by resection, liver transplantation or ablation, but many patients have a poor prognosis due to unresectable tumor. Medical therapy includes the multi-kinase inhibitors sorafenib and levatinib for unresectable HCC, but there are side-effects that impinge on the patient’s quality of life [8,57].

Since the majority of patients with HCC have underlying liver disease, most commonly cirrhosis, there has been some trepidation regarding the use of ICIs for HCC. ICIs have been investigated for treatment of HCC, as monotherapy or combination therapy with both locoregional treatment and other forms of chemotherapy [58]. Recently, the results of a phase 3 trial of atezolizumab, a PD-L1 inhibitor, together with bevacizumab, an inhibitor of angiogenesis via targeting of VGEF, has been published [59]. The results showed a superior overall and progression-free survival compared to sorafenib. The incidence of grade 3–4 hepatic adverse effects was between 1.2% and 3.6%. Thus, it seems that atezolizumab has a useful role in treating patients with HCC, despite these patients having cirrhosis.

## 10. ICIs and Liver Transplant Recipients

Patients undergoing organ transplants have an increased risk of developing malignancies, which is attributed to the need for long-term immunosuppressive treatment [60]. Patients who undergo liver transplantation due to hepatocellular carcinoma have a high risk of relapse (between 15% and 20%) [61]. ICIs have, however, been used as salvage therapy in selected transplant patients with recurrent HCC. A recent report summarizes the results of 14 cases of liver transplant patients treated with ICIs for different malignancies [62]. There were seven patients with HCC and two with fibrolamellar HCC. Six of the seven patients were treated with nivolumab and one with pembrolizumab. The six HCC patients treated with nivolumab did not develop graft rejection and neither did the sole patient who received pembrolizumab. Both patients with fibrolamellar HCC were treated with nivolumab and both developed fatal rejection. Further studies are needed on patients with less advanced disease, to examine the effect of other medications, the place for biopsy and the role of immunosuppression.

## 11. Clinical Management of Immune Check Point Inhibitors-Induced Hepatotoxicity

The development of hepatotoxicity related to ICIs results from an activation of immune responses. There are limited data available on which to base management decisions and current guidelines for treatment are based on expert opinion. A high degree of awareness of this entity is required from oncologists and other physicians involved in treating these patients, including primary care physicians.

Before making the decision to administer ICIs, an initial assessment is mandatory. This should include a revision of the medical history, looking for consumption of alcohol and herbal medicines. These may have changed since treatment with ICIs commenced. Furthermore, there may have been addition of other medications including over-the-counter products. In addition, repeat examination for chronic liver conditions including viral infections, NAFLD and autoimmune disease should be undertaken. ICI treatment is not contraindicated in patients who have chronic liver disease and do not appear to incur an increased risk for developing ICI hepatotoxicity [63,64,65]. Physical examination of the patient for any signs of advanced liver disease and repeat baseline blood tests and imaging are important. The blood tests should include in addition to routine blood count and liver biochemistry, a metabolic and lipid profile and antibody tests for HBV, HCV, HIV and cytomegalovirus. In addition, autoantibodies consistent with autoimmune hepatitis including anti-nuclear antibody (ANA), anti-smooth muscle antibody (ASMA), anti-mitochondrial antibody (AMA, anti- liver-kidney-microsome (LKM) and anti -swine antibody (SLA) should be determined [66].

In view of the possibility of reactivation of tuberculosis, serum screening should also be performed [67].

The European Association for the Study of Liver disease (EASL) guidelines for management of severe DILI or hepatotoxicity that is unresponsive to steroids note that liver biopsy may be necessary. We suggest that this may not be appropriate in patients with metastatic disease, except in carefully selected cases. The procedure is invasive, with a risk of hemorrhage and of limited therapeutic benefit, especially in terms of years of quality of life. Furthermore, there is not a typical finding on liver histology and the main contribution may be to exclude other causes of liver disease. There may be some hints on histology—those patients who were treated with anti-PD-1/PD-L1 ICIs have a more heterogeneous picture compared to those treated with anti-CTLA-4 agents [30]. Anti-PD1/PD-1 inhibitors demonstrate a lobular inflammation with CD4+/CD8+ T cells, while anti-CTL4 inhibitors may have central vein endotheliosis and non-necrotizing granulomas [30]. There have also been reports of biliary injury in three cases of steroid-resistant anti-PD1 hepatoxicity [25].

Modern imaging techniques including ultrasound, computed tomography (CT) scan and magnetic resonance imaging (MRI) and magnetic resonance cholangio pancreatography (MRCP) may aid in diagnosis. Severe ICI hepatotoxicity has periportal edema and lympho- adenopathy, while milder cases may have a normal appearance [68].

This approach is supported by a retrospective study of 21 cases of a total of 453 IRI hepatotoxicity patients who were all managed by the above approach without liver biopsy [42].

Recently, a systematic literature review and a proposed algorithm have been published [43]. This was based on 107 cases of ICI-related DILI, of which 83 (78%) were grade 3–4. In almost all (99%) of the cases, the ICI that was implicated was withheld. Of the cases, 86% received corticosteroid treatment. There was resolution of the hepatotoxicity in nearly all (98%) of the cases. The time to resolution ranged from 8 to 104 days. Corticosteroid therapy, however, may not always be required [66]. There are also recommendations from the Society for the Immunotherapy of Cancer Toxicity Management working Group [69]. The American Gastroenterological Association has also published guidelines on the diagnosis and treatment of ICI colitis and hepatitis [43].

Liver enzymes need to be checked at baseline and prior to each treatment cycle, since most patients are asymptomatic. The patient should be assessed for hepatic adverse effects and the treatment should be targeted at the organ system with the most severe involvement. The grade of hepatocytotoxicity is determined by the CTCAE classification. Liver biopsy is not essential for diagnosis in many cases. The need for a firm diagnosis, to exclude metastasis and to know the histologic severity, needs to be weighed against the cost and risks associated with an invasive procedure.

Treatment with the ICIs should be stopped. This is recommended to be temporary for grade 2 reactions and permanent for grades 3 and 4. There are, however, reports of ICI reintroduction following cases of grade 3–4 hepatotoxicity without recurrence [44]. The study by Pollack et al. [45] reported 21 out of 29 patients with ICI hepatotoxicity from combined anti-CTLA-4 and anti-PD1 who were successfully retreated. Of these, 14 of 19 cases had successful repeat administration after grade 3–4 hepatotoxicity.

The initial treatment, after cessation of the ICI, recommended by all guidelines, consists of corticosteroids. An analysis of the data from a recent systematic review concluded that nearly half of the 26 patients with grade 3–4 ICI hepatocytotoxicity improved without steroid treatment [34]. Routine steroid therapy is not recommended for grade 2 reactions. If there is no improvement after stopping ICI treatment, oral prednisone at a dose of 0.5–1.0 mg/kg/day orally should be given. For grade 4 reactions, methylprednisolone at a dose of 1–2 mg/kg/day is recommended, although 1 mg/kg/day is adequate for the majority of cases. The main reason for trying to avoid the concomitant use of corticosteroids is the increase in risk of infection [70]. Nakano et al. consider that Mycophenolate mofetil is a successful theraoy of corticosteroid-resistant immune checkpoint inhibitor-induced hepatitis [71]. Horvat et al. analysis on immune-related adverse events, in patients with melanoma treated with ipilimumab is the need for systemic immunosuppression [72]. It remains to be seen what the immunosuppressive effect of the steroids will be on the progression of the tumor [36]. There are two reports suggesting a milder prognosis for acute liver injury (greater than grade 2 severity) in 16 patients treated with anti-PD-1/PD-L-1 and anti-CTLA-4 antibodies, either alone or in combination [30]. Six (38%) of the patients had an improvement in the liver enzymes, and in two, a successful rechallenge with ICI therapy was performed. There is an additional report of 10 patients with melanoma who developed ICI hepatotoxicity from PD-1 and/or CTLA-4 inhibition [73]. The hepatotoxicity resolved in all cases, but half did not receive steroids. Other immunosuppressive agents were not administered. In addition, the hepatotoxicity resolved sooner in the patients who did not receive steroids (median 4.7 weeks) compared to a median of 8.6 weeks in those who received corticosteroids.

The optimal dose of steroid therapy is also not clear due to the absence of data from trials comparing dosage. The dose of 1 mg/kg of prednisone is based on the dose used in autoimmune hepatitis. A comparison of two groups of patients with ICI hepatotoxicity treated with either 50–60 mg of prednisone per day or 1 mg/kg/day, found that there was no benefit in terms of time to ALT recovery [42].

Other immunosuppressives have been used as initial treatment but in smaller numbers in steroid-refractory ICI hepatotoxicity. These include mycophenolate [47], azathioprine [49], cyclosporine [42], tacrolimus [74], infliximab [50], urso-deoxycholic acid [51], anti-thymocyte globulin [75], tocilizumab [76] and plasma exchange [69]. Infliximab is often used for the treatment of steroid-refractory immune colitis with no reports of hepatotoxicity [77], although dormant hepatitis B virus infection may be reactivated and needs to be screened for pre-treatment.

Not all patients will respond to initial treatment with corticosteroids for ICI hepatotoxicity. A recent review summarized 19 cases [30,78]. Not all the patients required treatment with another immunosuppressive, 13 received mycophenolate, of whom 4 also received anti-thymocyte globulin. Other treatments included azathioprine, tacrolimus, infliximab, cyclosporine and toclizumab. There have, however, been reports of an autoimmune hepatocellular reaction related to infliximab [79,80].

A suggested approach to the treatment of ICI hepatotoxicity is shown in Figure 3.

## 12. Patient Education

When starting patients on an ICI, it is critical to educate them on potential drug toxicities they may experience as well as to establish close communication with them, all toward the goal of helping patients manage any adverse events before symptoms become severe. Specifically, it is important to discuss potential on-target, drug-specific toxicities, counseling patients to be proactive, even before experiencing symptoms. The physician should advise the patients to alert the clinician if they experience gastrointestinal (GI) toxicities, even if their symptoms are mild. The clinician should discuss management strategies and potential drugs that can be provided if the GI symptoms are persistent.

## 13. Summary

ICIs are a new class of immune modulators that are being employed for an increasing number of advanced malignancies. They have many side-effects, including hepatotoxicity. Understanding the pathogenesis of liver injury may ultimately allow better and quicker diagnosis of this adverse reaction. Experience is growing on how to treat these side-effects and it may be possible in some cases to reintroduce the medications after an adverse effect.

This review described the scenario in which due to the ICI, all types of liver cells are sensitized. There is an overproduction of proinflammatory cytokines that will lead to an inflammation of hepatocytes. Other cytokines and chemokines such as Endothelin 1, RANTES, IL-10/IL-8 and leptin will contribute to the inflammatory microenvironment. Transforming growth factor beta (TGF-β) will transform the stellate cells in miofibroblasts that will produce an abundant extracellular matrix leading to fibrosis and cirrhosis.

## Figures and Tables

**Figure 1 biomedicines-09-00101-f001:**
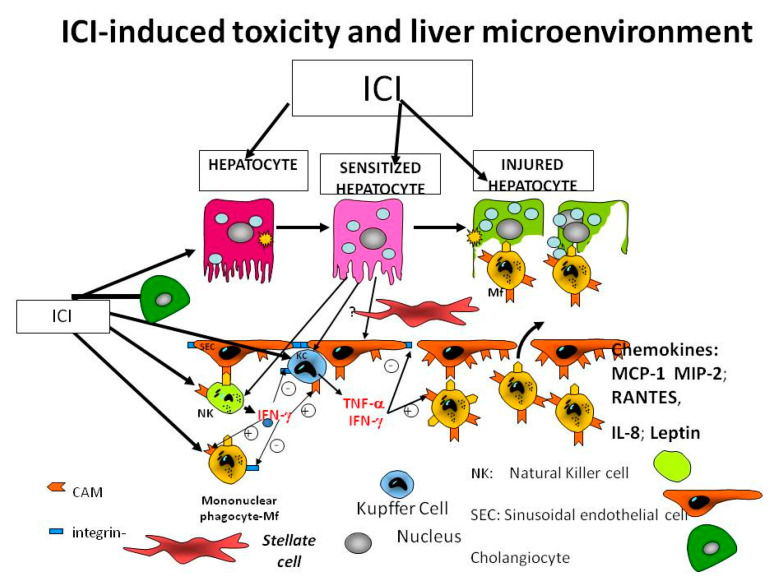
Liver environment is changed by the immune checkpoint inhibitors (ICI).

**Figure 2 biomedicines-09-00101-f002:**
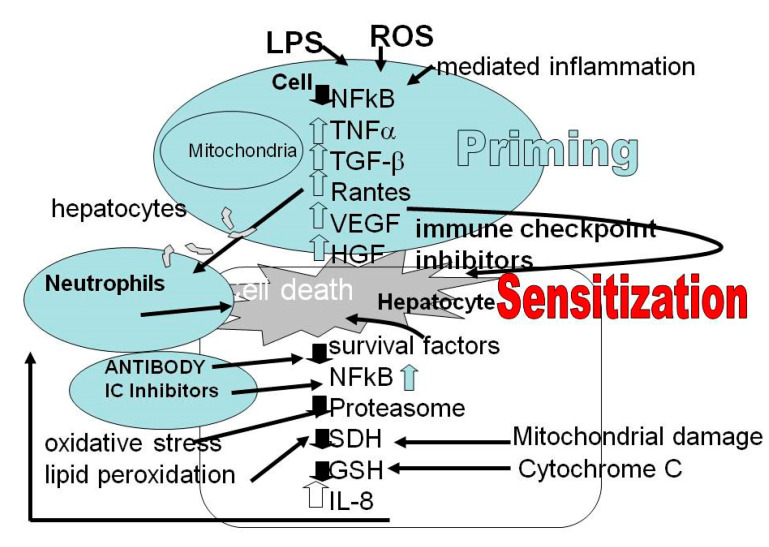
Hepatocyte reaction to signals of inflammation, oxidative stress, death and survival and the role of IC inhibitors. Reactive oxygen species (ROS), lipopolysaccharides (LPS), nuclear factor kB (NF-kB), Interleukin (IL: Il-1ß, Il-6), tumor necrosis factor alpha (TNF α), chemokines (IL8; RANTES CCL5: RANTES—Regulated upon Activation, Normal T cell Expressed and presumably Secreted), CC-chemokine ligand 2 (CCL2), specific mitochondrial enzyme succinate dehydrogenase (SDH) and release of cytochrome c, TGF-β—Transforming growth factor beta, VEGF—vascular endothelial growth factor, HGF—hepatocyte growth factor.

**Figure 3 biomedicines-09-00101-f003:**
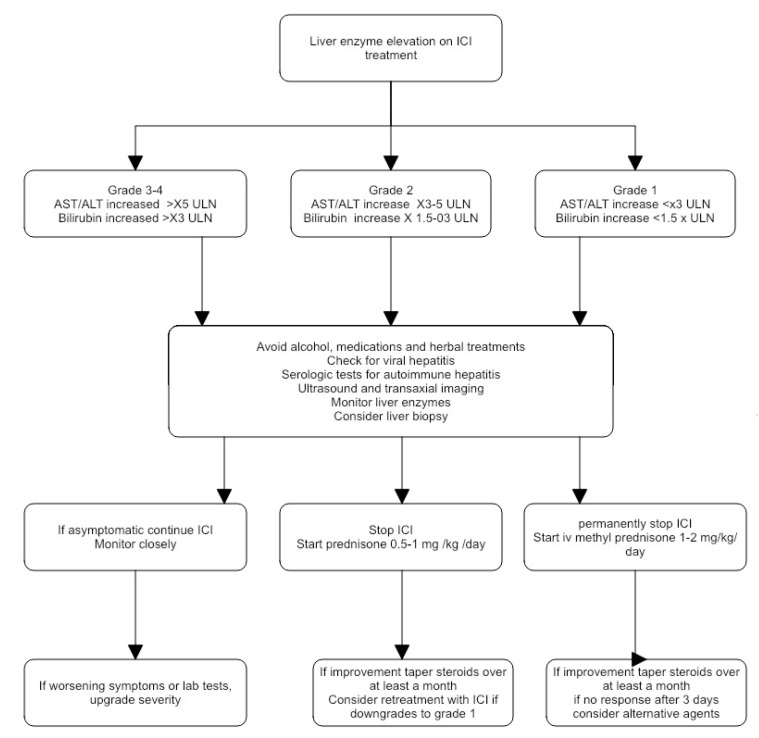
A suggested approach to the management of ICI hepatotoxicity.

**Table 1 biomedicines-09-00101-t001:** The differences between autoimmune hepatitis and ICI hepatotoxicity.

	Autoimmune Hepatitis	ICI Hepatotoxicity
Gender	Female predominant	Equal sex incidence
Symptoms	Malaise, jaundice	Fever, rash
Antibody—ANA, ASMA	Positive	Negative or low titer
Gamma globulin level	Increased	Normal range
Histology	Interface hepatitis, fibrosis	Hepatitis (lobular, pan-lobular, centrilobular, granulomatous). Portal fibrosis.
Cell infiltration	Plasma cell: CD4+ CD8+	Histiocyte: CD4+ CD8+
Recurrence after ICI withdrawal	Yes	No

ANA—antinuclear antibody, ASMA—anti-smooth-muscle antibody.

**Table 2 biomedicines-09-00101-t002:** Summarizes the reported hepatic side-effects of ICI therapy.

Ref	CPIH (N)	CPIH Severe (N)	CS Tx	CS Type	Duration CS Tx	Non-CS Tx	Time to Normal LFT	ALT (IU)	Serology	ICI	Liver Biopsy (N)
[42]	17	11 (8- G3/3-G4)	12	1 no CS 12 PR, 1 mg/kg/day 1 IV Dx 3 MPR 1 g/day	42 (7–78) day	1 Azathio-prine, 1 Cyclospo-rine	31 days	447 (59–2355)	N/A	Ipilimumab/ nivolumab/ Pembrolizumab/ indoximod/ Vemurafenib/ dabrafenib	N/A
[43]	10	9 (7-G3, 2-G4)	5	N/A	N/A	0	2–55 weeks	416 (155–1735)	ANA > 1:80 (1), AMA 1:1600 (1) PBC	anti-CTLA-4 (*n* = 6), Anti-PD-1/PD-L1 (*n* = 3), combination (*n* = 1)	2 granulomatous hepatitis associated with a moderate and polymorphous inflammatory infiltrate, no interface hepatitis
[44]	16	16	10	6 spontaneous improvement 7 oral CS 0.5–1 mg/kg/day; 2 oral CS 0.2 mg/kg/day 1 IV steroid 2.5 mg/kg/day	N/A	1 MMF	N/A	437 (147–2289)	ANA > 1:80 (8), ASMA > 1:80 (3)	Anti-CTLA-4 (n = 7) Anti-PD-1/PD-L1 (*n* = 9)	the portal inflamatory infiltrate contained numerous eosinophilic polynuclear cells
[44]	1	1		MPR 1 mg/kg/day then Steroid	152 days	UDCA	N/A	N/A	ANA 1:3 (20)	Nivolumab	moderate lymphocytic inflammatory infiltrate, bile duct injury; mild periportal necrosis; PD-L1 IHC, using anti-PD-L1/CD274 (clone SP142); a strong granular immuno-reactivity in the cytoplasm of Kupffer cells and hepatocytes.
[45]	29	19 (G3/4)	28	1 no CS 28 PR 0.5–1 mg/kg/day	35 (5–240)	3 MMF	N/A	N/A	N/A	Combination	N/A
[46]	21	14 (9-G3, 5-G4)	19	2 no CS; 11 Pr 7 MPR1 IV DX	N/A	8 MMF 1 Tacrolimus 1 Infliximab	112 days	732 (73–2857)	N/A	Combination	NO
[47]	1	1	1	MPR 500mg/day then PR 150 mg daily	9 days then 6 weeks	MMF anti-thymocyte globulin	37	1.peak 2521 2.peak 6362	negative	Ipilimumab	NO
[48]	1	1	1	MPR 2 mg/kg/day then PR	4 days then 6 weeks	MMF anti-thymocyte globulin	30	4700	Negative	Ipilimumab	NO
[49]	3	3	3	IV-MP 1g/kg then PR	3-day pulse then tapering	0	rapidly	886 (553–1211)	negative	Ipilimumab	NO
[50]	1	1	1	PR 1 mg/kg/day x 4 and 2 mg/kg/day	30 days	0	8 days	250	negative	Ipilimumab	N/A
[50]	1	1	1	PR 2 mg/kg/day	15 days	Artificial liver plasma ex-change	LFT did not improve	1269	0	Pembro lizumab	N/A
[51]	1	G 3	0	spontaneously recovered	N/A	N/A	N/A	N/A	N/A	Nivolumab	N/A
[52]	1	1	1	MPR 2 mg/kg/day and pulse therapy	N/A	Azathioprine	30	539	0	Nivolumab	N/A
[53]	1	1	1	MPR 2 mg/kg/day 10 days, PR 1 g/kg/day with tapering	~90 days (all Cs tx with tape- ring)	MMF anti-thymocyte globulin	27	1900	0	Ipilimumab & nivolumab	N/A
[54]	1	1	1	MPR pulse for 6 day; MPR 1 g/kg/day then oral PR 1.25 mg/kg/day	6-day pulse then tape ring	MMF	104	1623	0	Ipilimumab	N/A
[55]	1	G 4	1	Oral MPR 0.6 mg/kg/day; half-pulse 500 mg/day	N/A	UDCA	4 months after end of nivolu mab	693	0	Nivolumab	Portal area with inflammatory cells, including lymphocytes and eosinophils.
[55]	1	G 4	1	MPR 2 mg/kg/day for 4 days; then, DX, 3 days MPR 1 g/day, followed by PR 150 mg	7 days then tape ring	MMF anti- thymocyte globulin	Persis- ted with grade 1–2 CPIH	~1250	ASMS 1:1 (60)	anti-PD-1	inflammatory infiltrate around the portal tracts and central veins, focal necrosis. PD-L1 was expressed on hepatocytes; in the infiltrating lymphocytes, PD-1 was expressed at low levels
[55]	1	1	1	MPR 2 mg/kg/day, then 4 & 6 mg/kg/day	14 days	MMF	55 days	~350	Nega- tive	nivolumab	Inflammation; eosinophilic and neutrophilic granulocytes; perivenular (zone 3) cholestasis.

ALT—alanine aminotransferase; ASMA-anti-smooth muscle antibody; DX-oral dexamethasone; G-grade; ICI—immune checkpoint inhibitor; CPIH—Check point inhibitor hepatotoxicity; CS—corticosteroid; CS Tx—Corticosteroid therapy; IV—Intravenous; LFT—liver function test; MMF—Mycophenolate; MPR-Methylprednisolone; UDCA—urso-deoxycholic acid; Tx—Therapeutic agent.

## Data Availability

All the authors have the data available in any time.

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
