# Peer review of "Checkpoint Inhibitors and Hepatotoxicity"

_biomedicines, 2021, doi:10.3390/biomedicines9020101_

Round 1

Reviewer 1 Report

The manuscript is improved and is now more focused on hepatotoxicity. I think it is just acceptable, although novelty is limited.
The abstract should be rewritten also, to strictly focus on hepatotoxicity of ICI. The section on biomarkers should only focus on potential biomarkes of hepatotoxicity - tumor biomarkers are out of the scope.

Author Response

Thank you very much for the suggestions. We performed the requested changes.

Reviewer 2 Report

No further comments

Author Response

Thank you very much for the  recommendation. We checked again the manuscript and performed the requested changes

This manuscript is a resubmission of an earlier submission. The following is a list of the peer review reports and author responses from that submission.

Round 1

Reviewer 1 Report

Stephen Malnick et al. performed a moderate revision of the manuscript, particularly they corrected some errors indicated in previous comments. The authors have prepared two more figures generally more appropriate to the content of the manuscript. However, the revision did not improve much the overall quality of the manuscript. Particularly, authors failed managing abbreviations (this problem was indicated in reviewer’s comments). Still, there is confusion with their use in the revised manuscript: some of them are deciphered twice (TME, lines 38 and 65; TIM-3, lines 78 and 83; RUCAM, lines 208 and 241), others should not de introduced since they are used only ones in the manuscript (PDGF, EGF, CAMs, VCAM-1 etc.). The new “Microenvironment” section looks not logical at the beginning of the manuscript devoted to hepatotoxic effects of immune checkpoint blockade and needs some links to the following text. In this section the checkpoint molecules LAG-3 and TIM-3 suddenly arise, whereas PD-1 is mentioned in passing and CTLA-4 not mentioned at all. Figure 1 is good but not described properly and looks to be in wrong place within the manuscript. Some inappropriateness can be found throughout the whole text.

So, I have to repeat my remarks: the text is written superficially and carelessly.

I do not recommend this manuscript in its present form for the publication in Biomedicines.

Author Response

1-Stephen Malnick et al. performed a moderate revision of the manuscript, particularly they corrected some errors indicated in previous comments. The authors have prepared two more figures generally more appropriate to the content of the manuscript. However, the revision did not improve much the overall quality of the manuscript. Particularly, authors failed managing abbreviations (this problem was indicated in reviewer’s comments). Still, there is confusion with their use in the revised manuscript: some of them are deciphered twice (TME, lines 38 and 65; TIM-3, lines 78 and 83; RUCAM, lines 208 and 241), others should not de introduced since they are used only ones in the manuscript (PDGF, EGF, CAMs, VCAM-1 etc.). 

Answer: We delete the duplicates and we did not introduce   PDGF, EGF, CAMs, VCAM-1

2-The new “Microenvironment” section looks not logical at the beginning of the manuscript devoted to hepatotoxic effects of immune checkpoint blockade and needs some links to the following text.

2-Answer:The reviewer asked for the “Microenvironment” section in his first review of our work. We consider that we choose the correct place to introduce the concept.

3-In this section the checkpoint molecules LAG-3 and TIM-3 suddenly arise, whereas PD-1 is mentioned in passing and CTLA-4 not mentioned at all.

3-Answer: Cytotoxic T -lymphocytes associated antigen 4 (CTL-4) was introduced.

4-Figure 1 is good but not described properly and looks to be in wrong place within the manuscript.

4-Answer.

The explanation is properly described.

Reviewer 2 Report

This is an interesting review on hepatotoxic effects of ICI. I have just some few comments:

  • the authors should include a dedicated section on clinical management of ICI-toxicity (diagnosis, treatment, guidelines)
  • the authors should further discuss the difference between mono-therapies and combinations

Author Response

Thank you very much for the review. We appreciate the suggestion.

We introduced the required section on clinical management 

Reviewer 3 Report

Being the cause of significant morbidity and even death, checkpoint inhibitor-induced hepatotoxicity is of high clinical interest. This is reflected by several recent and excellent reviews published on the topic, including Peeraphatdit et al., Hepathology 2020, and Lleo et al, Digestive and Liver Disease 2019, 55:1074-1078.

To my opinion the current review does not add significant biological background knowledge or guidance to clinicians to that of the abovementioned reviews and several clinical guidelines. Hence, the authors might chose to approach the topic from some different angle to increase the interest in and novelty of the work. 

A very large part of the paper is a detailed review of general biological mechanisms of cancer immunology, and in fact hepatoxicity is first discussed from line 200. This general part is nicely outlined, but not directly relevant to the topic and could be shortened considerably.

In Figure 3 the column describes serious toxicity on the left side in the first line of boxes, but continues on the right in the lower two lines of boxes, which is confusing. 

Author Response

1-Being the cause of significant morbidity and even death, checkpoint inhibitor-induced hepatotoxicity is of high clinical interest. This is reflected by several recent and excellent reviews published on the topic, including Peeraphatdit et al., Hepathology 2020, and Lleo et al, Digestive and Liver Disease 2019, 55:1074-1078.

1- Answer

We cited and discussed the paper.

2-To my opinion the current review does not add significant biological background knowledge or guidance to clinicians to that of the abovementioned reviews and several clinical guidelines. Hence, the authors might chose to approach the topic from some different angle to increase the interest in and novelty of the work.

 2-Answer

The paper add significant biological background knowledge or guidance to clinicians.

3-A very large part of the paper is a detailed review of general biological mechanisms of cancer immunology, and in fact hepatotoxicity is first discussed from line 200. This general part is nicely outlined, but not directly relevant to the topic and could be shortened considerably.In Figure 3 the column describes serious toxicity on the left side in the first line of boxes, but continues on the right in the lower two lines of boxes, which is confusing. 

3-We consider that the review brings new mechanistic and medical details that help the audience to understand the magnitude of the problem. Moreover we added an entire section on possible medical clinical management of the hepatotoxicity produced by ICI in certain individuals.

Round 2

Reviewer 1 Report

The authors have done some work to improve the quality of the manuscript. In particular, in most cases, they sorted out the abbreviations and improved the presentation style.

However, a number of remarks remain. First of all, this concerns some confusion of the presentation. Apparently, the main problem of the authors is not in the level of English, but in the fact that they do not clearly understand what they want to say in some parts of the manuscript. Let me explain what I mean by the example of the first part.

It is not clear why CAR T lymphocytes are mentioned in the abstract, although they do not have a word in the manuscript. The very essence of the work should be stated in the abstract. If desired, authors may place CAR T lymphocytes in the introduction.

1) The very first sentence of the introduction suggests that the authors do not fully understand the terminology they use. Immune checkpoint inhibitors are most often antibodies, but not always antibodies. Formally it can be fragments of antibodies, other protein scaffolds, chemical inhibitors…

2) Next, the authors write (lines 44-45) about the use of immune checkpoint inhibitors “to stimulate the immune systems ... in order to reactivate the immune system ...”. At this point, authors need to correct the style by removing this semantic repetition.

3) The next sentence (lines 45-46) can be understood to mean that antibody checkpoint blockade acts as a regulator of excessive inflammation. But the regulation is provided not by antibodies, but by the checkpoints themselves when interacting with the corresponding ligands, and the blockade cancels their regulatory action.The sentence needs to be rewritten.

4) Then there are a couple of sentences in which the basic concept is presented illogically (lines 46-48). The authors state that the expression of checkpoints increases in tumor cells, the ligands of which inactivate T cells. This fragment needs to be reformulated by correcting the logic.

5) Lines 59-66 in this paragraph essentially says the same thing twice. This unnecessary semantic repetition should be removed.

6) In the next paragraph (line 67), in fact, there are also repetitions,  this paragraph can be significantly reduced.

It contains expression “intrinsic and extrinsic elements of the immune system”, it is not clear which elements the authors classify as intrinsic, and which ones as extrinsic, this needs to be written in more detail.

7) At the end of the section “Tumor microenvironment” it is worth listing and BRIEFlY describing the most famous immune checkpoints, including PD-1, CTLA-4, LAG-3, TIM-3. This will create a logical link to the main topic of the review.

The title of the paragraph Biomarkers needs to be clarified - biomarkers of what? Titles of other paragrapfs should be critically reconsidered.

Again there are strange decryptions of abbreviations, such as protein IK3 oncogene (PIK3CA), MYC -Oncogene c-MYC.

Figure 2 in the body of the manuscript is poorly described.

The fact that the main purpose of the review was formulated only at the end of paragraph 4 suggests that paragraphs 1-4 are redundant. They need to be shortened, or the purpose of the review should be reformulated.

The second part of the review is more meaningful and better written.

And finally, the authors' extraordinary carelessness should be noted. With their great efforts, which, judging by the text, were applied to correct the manuscript, they could not even check the abstract thoroughly - in the middle of the sentence on the second line, a capital letter A suddenly appears. In the next section, which I analyzed in detail, there are dots put in an unnecessary place (line 48), the rest of the square brackets (line 61), extra letters (line 87). It should be noted that «beta» is not put after the TGF in some cases (lines 145, 166). This does not improve the impression of this work in any way. I am writing about the need to carefully read the manuscript and correct such minor flaws for the third time and hopefully for the last time.

In general, I do not consider it my task to directly indicate all those places where the manuscript needs to be corrected in order to improve it. This is the task on the shoulders of the authors.

Reviewer 3 Report

The authors intend to focus on CPI and hepatotoxicity, but this this not at all reflected in the abstract and the very long introduction also focus only on general cancer immunology. Indeed, on line 184 hepatotoxicity is mentioned for the first time! It is not clear to me that the manuscript brings significant knowledge to that already published in recent reviews on the same topic.